# An Integrated Centrifugal Degassed PDMS-Based Microfluidic Device for Serial Dilution

**DOI:** 10.3390/mi12050482

**Published:** 2021-04-23

**Authors:** Anyang Wang, Samaneh Moghadasi Boroujeni, Philip J. Schneider, Liam B. Christie, Kyle A. Mancuso, Stelios T. Andreadis, Kwang W. Oh

**Affiliations:** 1Sensors and MicroActuators Learning Lab (SMALL), Department of Electrical Engineering, The State University of New York at Buffalo, Buffalo, NY 14260, USA; anyangwa@buffalo.edu (A.W.); philschn@buffalo.edu (P.J.S.); liamchri@buffalo.edu (L.B.C.); kylemanc@buffalo.edu (K.A.M.); 2Department of Chemical and Biological Engineering, University at Buffalo, State University of New York (SUNY-Buffalo), Buffalo, NY 14260, USA; samanehm@buffalo.edu (S.M.B.); sandread@buffalo.edu (S.T.A.); 3Department of Biomedical Engineering, University at Buffalo, State University of New York (SUNY-Buffalo), Buffalo, NY 14260, USA; 4Center of Excellence in Bioinformatics and Life Sciences, University at Buffalo, State University of New York (SUNY-Buffalo), Buffalo, NY 14263, USA

**Keywords:** microfluidic, vacuum pumping, centrifugal pumping, serial dilution, qPCR, PDMS

## Abstract

We propose an integrated serial dilution generator utilizing centrifugal force with a degassed polydimethylsiloxane (PDMS) microfluidic device. Using gas-soluble PDMS as a centrifugal microfluidic device material, the sample can be dragged in any arbitrary direction using vacuum-driven force, as opposed to in a single direction, without adding further actuation components. The vacuum-driven force allows the device to avoid the formation of air bubbles and exhibit high tolerance in the surface condition. The device was then used for sample metering and sample transferring. In addition, centrifugal force was used for sample loading and sample mixing. In this study, a series of ten-fold serial dilutions ranging from $10^{0}$ to $10^{-4}$ with about 8 μL in each chamber was achieved, while the serial dilution ratio and chamber volume could easily be altered by changing the geometrical designs of the device. As a proof of concept of our hybrid approach with the centrifugal and vacuum-driven forces, ten-fold serial dilutions of a cDNA (complementary DNA) sample were prepared using the device. Then, the diluted samples were collected by fine needles and subject to a quantitative polymerase chain reaction (qPCR), and the results were found to be in good agreement with those for samples prepared by manual pipetting.

## 1. Introduction

Strategies for the generation of bio-molecular and chemical concentration gradients have been widely studied in the context of applications involving sample preparation and analysis. Examples include cell migration for chemotaxis [1], quantitative polymerase chain reaction (qPCR) [2,3], enzyme-linked immunosorbent assay (ELISA) [4], and high-throughput screening (HTS) [5]. Various approaches are used to generate serial dilutions of liquid samples in biology and chemical research.

Microplates are widely used in the laboratory research setting as a standard tool in sample preparation and analytical quantification. Each “well” or small fluid reservoir typically has a capacity in the range of nanoliters to microliters. Furthermore, each plate can have up to thousands of sample wells. For instance, one of the most common microplate formats is the 96-well microplate with a capacity of 400 μL/well. A typical procedure involves manual pipetting of samples into each microwell and mixing with dilution buffers. This is a time-consuming process that requires some form of a skilled operator. In addition, microplates with more than hundreds of wells are not practical for manual operation. Recently, several companies have developed liquid handling robots in an attempt to automate the sample preparation process [6,7]. Robotic systems can mimic tedious tasks with a high rate of accuracy and precision (e.g., Tecan system [8]). Basic robots can achieve simple dispensing of an allotted volume of sample, while more advanced robots can manipulate the position of pipettes and microplates to achieve automated laboratory sample preparation and analysis. Although such robotic systems are critical for high-throughput applications, their implementation is costly and requires complementing infrastructure.

A lab-on-a-chip (LOC) integrates multiple laboratory functions into a single platform. Some widely used functions include sample preparation [1,2,9], high-throughput screening [5], virus detection [10], and point-of-care testing (POCT) [11,12,13,14]. Offering an alternative to using systems requiring time-consuming manual operation or costly robot-based liquid handling, microfluidic platforms are emerging as an inexpensive solution for applications in which the automation of sample preparation and analysis schemes are necessary. Furthermore, LOC and microfluidic technology represent a useful strategy for generating concentration gradients (e.g., serial dilution) [15,16,17,18].

Lee et al. [15,19] developed a pressure-driven dilution device for high-throughput screening. Three additive samples with a buffer solution are mixed and diluted inside the device to generate seven combinations of liquid solution. Mixing and dilution ratios of outcome solutions are designed based on a fluidic network study. The disadvantage of this approach is that it requires multiple bulky mechanical pumps to operate the device. Gao et al. [20] proposed a microfluidic gradient generator for chemotaxis using passive pumping. Operating the device without bulky external actuation peripherals is highly useful for end-users, such as researchers and bedside patients. In addition to end-users saving on the costs of introducing expensive peripherals, such passive pumping is simple to operate. Examples are capillary microfluidics [14], paper-based microfluidic [21], and vacuum-driven microfluidics [13]. However, achieving concentration gradients through passive pumping is challenging because of the difficulty in controlling the inherent pressure difference between liquid samples [22,23]. This pressure difference could arise from mismatches in material properties, liquid properties, inlet port size, and loaded sample volume. Strategies to offset this inherent pressure difference, i.e., via a pressure balancer, to achieve a concentration gradient, are attracting widespread attention. Several reported approaches include ladder structure [20,24], bridge over the inlets [25], and oil sealing [26]. Another challenge to generating concentration gradients in passive pumping is its limited flow rates owing to the reliance on surface tension [14] and material properties [11], resulting in lengthy operation times.

Among various approaches, the preparation of serial dilutions using centrifugal forces has rarely been studied. Recently, Peter et al. [2] reported the automated preparation of serial dilutions using fill-level-coupled valves in a centrifugal microfluidic device. In their study, the valves were actuated via temperature change rate (TCR)-controlled valves to generate a series of ten-fold serial dilutions ranging from $10^{0}$ to $10^{-5}$. However, this technique requires additional temperature-controlling functionality and equipment for device operation. Centrifugal forces can only deliver the sample in a radially outward direction, which is considered a major disadvantage for integrated and automated workflows. To extend the capability in centrifugal microfluidics, additional actuation methods are used together to control the sample into desired directions for sample metering, sample transfer, and sample mixing [27]. Clime et al. [28] reported a combination of centrifugal and pressure-driven microfluidics. A microfluidic device is connected to pneumatic tubing, where a supplementary pneumatic controlling system is built into the rotating platform. The device can conduct liquid sample mixing and valving, suitable for multi-step workflows. However, the requirement of additional peripherals, such as electronic control and pneumatic pumping systems cannot be avoided. Strohmeier et al. [3] reported a fully integrated dilution series generation on a laboratory centrifuge using siphon valves. The reported device achieved a ten-fold dilution with 30 μL volume in five discrete dilutions of reagent. The limitation of the device is that the device performance was surface-sensitive, and the channel was required to be hydrophilic to utilize the siphon valves effectively.

Vacuum-driven pumping has been attracting increased attention as an alternative to passive pumping methods [13,29,30]. Gas-solubility-based vacuum-driven pumping utilizes the gas solubility property of materials (Figure 1). The device is made from materials with good gas solubility is degassed before use. Advantages of the vacuum-driven approach are absence of bubble formation [11,31,32,33], high tolerance to surface conditions [13,29], and robust passive pumping [11,13]. Vacuum-driven pumping can easily be integrated with other types of pumping without the need for additional peripherals.

This paper proposes an integrated centrifugal, degassed, Polydimethylsiloxane (PDMS)-based microfluidic device for serial dilution applications. The device utilizes both centrifugal force and vacuum-driven force to drive a liquid sample with the assistance of geometrical constraints, such as capillary valves [14,34,35] and phase-guide structures [36]. The use of a combination of two actuation methods complements the drawbacks of each. Limitations of centrifugal pumping, such as one-directional force and air bubble generation, can be overcome by introducing vacuum-driven pumping in combination. Limitations of vacuum-driven pumping, such as restricted flow rates and difficulty in achieving sample mixing because of the inherent pressure difference, can be compensated by using this method in combination with centrifugal microfluidics.

## 2. Theory and Working Principle

### 2.1. Switch a Dominant Force between Centrifugal Force and Vacuum-Driven Force

Centrifugal force and vacuum-driven force are the two main forces being leveraged in this device, as shown in Figure 2. This is aside from the effects of a Coriolis force and the moment of inertia. The dominant force is switched between the centrifugal force and the vacuum-driven force during device operation. The conditions to trigger centrifugal flows and vacuum-driven flows are compared and illustrated in Figure 2b,c. The centrifugal flows require a spin with a rotational speed $N_{\mathrm{RPM}}$ and a vent to deliver the liquid sample into the chamber. The vacuum-driven flows require the device made from materials with good gas solubility (e.g., PDMS) to be degassed before operation. Additionally, the chamber should be dead-end to allow liquid sample delivery. For example, in Figure 2b, the sample can be loaded from the top chamber to the bottom chamber, where the vent is open to the air, by applying angular velocity (centrifugal force). Once the sample was transferred to the bottom chamber, the sample from the chamber to the adjacent chamber, there is no opening and considered as a dead-end chamber. Considering the vacuum-driven pumping concept in Figure 2c, the sample would be transferred by vacuum-driven pumping. In short, to drive the sample with a centrifugal force, the target chamber should be open to the air in addition to the angular velocity, while to drive the sample with vacuum-driven force, the target chamber should be close. If wisely designed the device and let the liquid sample go through the open chamber and the close chamber one after another, we can make the situation to drive the sample using the centrifugal force and the vacuum-driven force one after another.

### 2.2. Device Structure

Figure 3 shows the images and schematics of the proposed device for a ten-fold serial dilution. The device has two inlets: inlet A for sample loading and inlet B for buffer loading. There are five sets of metering chambers and dilution chambers corresponding to serial dilutions with concentrations of $10^{0},10^{-1},10^{-2},10^{-3}$, and $10^{-4}$, aside from waste chambers for holding the excess volume of sample and buffer after the metering process. The device has three vents: two vents are in waste chambers, and one vent is in #1D (the ports of other vents in #2D, #3D, and #4D are not punched in this study). First, metering chambers are designed to allocate 8.96 μL in #1 M; 8.07 μL each in #2M, #3M, and #4M; and 7.26 μL in #5M. Dilution chambers are constructed with the aim of having a series of ten-fold serial dilutions ranging from $10^{0}\;\mathrm{to}\;10^{-4}$. The device is composed of three layers: a PDMS top layer, a PDMS bottom layer, and a glass slide substrate. The bottom layer possesses a symmetric design concerning the top layer to possess sufficient chamber height for efficient mixing and relatively large sample volumes (~7 to ~10 μL). A single layer of PDMS microfluidic channel/chamber with a glass slide substrate may work as well; however, it is not easy to keep uniform, especially concerning the depth of chambers of a few hundred micrometers scale. The fabrication process will be discussed later. However, making thick chambers in a single layer of PDMS using the soft lithography process is not easy and can be subject to the edge bead effect [37], resulting in a non-uniform depth.

### 2.3. Working Principle

To generate a series of ten-fold serial dilutions, the sample is subject to the metering process, transfer from the metering chambers to the dilution chambers, mixing in each dilution chamber, and transfer of one-tenth of the sample volume from one dilution chamber to the next dilution chamber. Figure 4 shows the core working principle of our proposed device using both the centrifugal and vacuum-driven forces. Before usage, all PDMS devices were degassed in a vacuum chamber under ~0.1 ATM overnight. (i) First, the sample and buffer were loaded into the device using a pipette. (ii) The liquid sample and buffer were metered by vacuum-driven flows: 8.96 μL (#1) for the sample containing blue dye; 8.07 μL (#2, #3, and #4) and 7.26 μL (#5) for the buffer containing red dye. Here, in order to achieve the same sample volume (~8 μL) of the dilution series, in the end, the first chamber has ~10% more volume, and the last chamber has ~10% less volume than the targeting sample volume in the metering process, considering the serial dilution process. Abrupt geometrical height/width differences in channels and chambers using phase-guide structures [36] enabled precise metering and aliquoting of the designated volume of liquids in each metering chamber [5]. (iii) Centrifugal force with 1400 RPM was applied to transfer liquids into each dilution chamber. Due to the Coriolis force, the sample transferred in the dilution chambers may not have a flat interface (see Appendix A). If needed, abrupt angular acceleration, deacceleration, or mechanical vibrations may be applied for smooth transition to the next process. With this RPM settings, there were always residues trapped in the capillary bridges and edges of the metering chambers. These trapped samples partially contribute to form a dead-end chamber, which can trigger vacuum-driven force to convey the sample in the later process. The trapped sample volumes had up to ~0.2 μL each (see Appendix A). (iv) Loading ports were sealed with adhesive tape (e.g., 3M tape) to avoid contamination and sample evaporation. As in the previous process, the dead-end chamber was formed. Thus the sample can be triggered by a vacuum-driven force. One-tenth of its original liquid volume stored in the dilution chamber was metered with the assistance of phase-guide structures and transferred to the next dilution chamber across the transfer bridge by vacuum-driven pumping. This enables the concentration of the sample to be reduced by a factor of ten: that is, to one-tenth of the original concentration. (v) and (vi) Centrifugal force was applied to mix the sample with the buffer. (vii) This process was repeated four times to achieve a series of ten-fold serial dilutions ranging from $10^{0}$ to $10^{-4}$ in each dilution chamber. (viii) and (viii’) The serial dilution process was completed. The images show the completed serial dilution with and without tapes.

### 2.4. Metering of Liquids

Metering of liquids is achieved by vacuum-driven pumping and phase-guide structures (Figure 5 and Appendix A). First, the sample was loaded into inlet A and buffer into inlet B by manual pipetting. Gentle pressure was applied while loading the liquids, causing the sample or buffer to fill the whole metering chambers and the vents. Then, the liquid sample or buffer would stop at capillary valves but keep flowing into the dead-end waste chamber because of the vacuum-driven pumping in the degassed PDMS device. Here, the strong constraint at the phase-guide structure causes the liquid column to separate into two parts: one flowing into the waste chamber and the other retained inside the metering chamber.

Various papers report several capillary valves, such as burst valves, retention valves, and triggering valves, for controlling a sample injection sequence in capillary microfluidics [34,35]. In addition, our previous study [38] reported sequential injection in vacuum-driven microfluidics when a device had multiple inlets and vents connected with capillary valves. The triggering order could be well controlled by designing different geometric structures for each capillary valve. In this study, we designed a degassed PDMS device in which the capillary pressure across the capillary valve next to the vent with a rigid neck shape, $\Delta P_{\mathrm{cV}}$, was always higher than that next to the inlet with a loose neck shape, $\Delta P_{\mathrm{cI}}$. As a result, the vacuum-driven pumping would (1) first split the liquid at the phase-guide structure; (2) next, drag the detached liquid from the inlet and then from the vent, in sequence; and (3) finally, withdraw the liquid into the dead-end waste chamber.

### 2.5. Transfer of Metered Liquids into Dilution Chambers

Centrifugal force was applied to convey metered liquids into each dilution chamber. To overcome the capillary pressure across the capillary valve placed between the metering chamber and the dilution chamber, the device was spun at an angular velocity, ω, higher than the burst angular velocity, $\omega_{b}:\;\omega >\omega_{b}$. The driving pressure due to the centrifugal force, $\Delta P_{d}$, exerted on the liquid column, whose advancing front-end and receding rear-end are located at distances $r_{A}$ and $r_{R}$ from the rotation center, can be described as below [39]:(1)$$\Delta P_{d}=\frac{1}{2}\rho \omega^{2}\left(r_{A}^{2}-r_{R}^{2}\right),$$
where *ρ* is the density of the liquid and ω is the angular velocity. In addition, the capillary pressure difference, $\Delta P_{c}$, between the advancing front-end and the receding rear-end of the liquid under equilibrium can be described by the Young-Laplace equation, as below [39,40]:(2)$$\Delta P_{c}=\Delta P_{\mathrm{cA}}-\Delta P_{\mathrm{cR}},$$
(3)$$\Delta P_{\mathrm{cA}}=-\gamma \;\left(\frac{\cos\theta_{\mathrm{tA}}^{*}}{h}+\frac{\cos\theta_{\mathrm{bA}}^{*}}{h}+\frac{\cos\theta_{\mathrm{lA}}^{*}}{w}+\frac{\cos\theta_{\mathrm{rA}}^{*}}{w}\right),$$
(4)$$\Delta P_{\mathrm{cR}}=-\gamma \;\left(\frac{\cos\theta_{\mathrm{tR}}}{H}+\frac{\cos\theta_{\mathrm{bR}}}{H}+\frac{\cos\theta_{\mathrm{lR}}}{W}+\frac{\cos\theta_{\mathrm{rR}}}{W}\right),$$
where $\Delta P_{\mathrm{cA}}$ and $\Delta P_{\mathrm{cR}}$ are the capillary pressure at the advancing front-end and at the receding rear-end. *γ* is the surface tension. $\theta_{\mathrm{tA}}^{*},\;\theta_{\mathrm{bA}}^{*},\;\theta_{\mathrm{lA}}^{*},$ and $\theta_{\mathrm{rA}}^{*}$ are the contact angles of top, bottom, left, and right considering the expansion angle of the channel at the advancing front-end. $\theta_{\mathrm{tR}},\theta_{\mathrm{bR}},\theta_{\mathrm{lR}},$ and $\theta_{\mathrm{rR}}$ are the contact angles of top, bottom, left, and right of the channel at the receding rear-end, respectively [39]. *h* and *w* are the height and the width of the channel at the advancing front-end. *H* and *W* are the height and the width of the channel at the receding rear-end.

When the centrifugal driving pressure is higher than the capillary pressure difference, or the driving angular velocity is higher than the burst angular velocity, the liquid can overcome the capillary barrier and, thus, transfer to the dilution chamber. The driving and burst angular velocity, ω and $\omega_{b}$, can be calculated by:(5)$$\Delta P_{d}\;\left(at\;\omega \right)>\Delta P_{d}\;\left(at\;\omega_{b}\right)=\Delta P_{c}.$$

In our design, when $\Delta P_{c}≅2186$ Pa, the corresponding burst angular velocity was $\omega_{b}≅131$
${\mathrm{rad}\;s}^{-1}$, or 1248 RPM. During operation, we applied ω = 1400 RPM counter-clockwise to ensure the loading of each metered liquid into each dilution chamber. Under experimental conditions, the residue was found to adhere to the edge parts of the metering chambers and the capillary valves; this is discussed later in the results and discussions section.

### 2.6. Partial Transfer of Liquids across the Transfer Bridge

Liquid transfer between neighboring chambers across the transfer bridge was achieved by vacuum-driven pumping with the assistance of phase-guide structures (Appendix A). The trapped sample at capillary bridges between metering chambers and dilution chambers assisted in forming a dead-end, while the sample was transferring from confined phase-guide structures into the next dilution chambers. Here, the trapped sample at capillary bridges was controlled. This is discussed in the later section. The phase-guide structures would determine the volume of the liquid to be transferred. Here, the phase-guide structures were designed to meter and transfer one-tenth of its original volume from the previous dilution chamber into the next dilution chamber. For a ten-fold serial dilution, one part of the sample from the previous dilution chamber was mixed with nine parts of the buffer in the next dilution chamber.

It should be mentioned that the liquid flowing into the dilution chamber from the metering chamber has a trajectory aimed at the left-side wall of the dilution chamber because of the effect of the Coriolis force (Appendix A) [41]. Therefore, the liquid interface should be made parallel to the phase-guide structures placed in the middle of the dilution chamber. This is a critical step to further transfer an exact volume of the liquid across the transfer bridge by vacuum-driven flows. Before the partial transfer, an additional spin with a rapid acceleration-and-deacceleration or mechanical vibration, or both, may help not only to achieve a straight liquid interface in contact with the phase-guide structures but also to mix the liquid sample with the buffer better.

### 2.7. Sample Mixing

Sample mixing is achieved with a partial contribution from the vortical flow, which leads to mixing in a micro-chamber due to Euler/inertial force from a continuous cyclic acceleration-and-deceleration under unidirectional rotation [42,43,44]. As reported by Ren et al. [43,44], a larger chamber, achieved by increasing the outer radius, height, or angular span of the chamber; a maximum rotation speed; and/or rapid acceleration-and-deacceleration would help to improve the mixing efficiency. For instance, the dilution chambers must be sufficiently thick to ensure efficient mixing. In our design, we used a chamber thickness of 420 μm to achieve mixing in less than a few minutes and to handle relatively large liquid volumes of ~10 μL. In the device, the thickness of phase-guide structures was optimized to be 210 μm. In the operation procedure, a 1400 RPM counter-clockwise rotation with up to 300 RPM/s cyclic acceleration and deacceleration for up to 3 min was applied to achieve uniform mixing.

## 3. Materials and Methods

### 3.1. Device Fabrication

The device was fabricated using soft lithography to construct microfluidic channels, chambers, and phase-guide structures in PDMS. Two wafers were used to fabricate molds for the top layer and the bottom layer of the PDMS device, having symmetrical patterns, as shown in Figure 1. The height of the channels and chambers was 210 μm, and the height of the phase-guide structures was 105 μm in each layer. First, molds were fabricated using a 3-inch silicon wafer (University Wafers, South Boston, MA, USA). The wafers were submerged into buffered hydrofluoric acid (BHF) at room temperature for 5 min to remove the native silicon dioxide layers. Next, the wafers were cleaned using acetone, then methanol, and finally rinsed with deionized water. Nitrogen blowing was then used to dry the surface.

The wafers were placed on a hot plate at 120 °C for 5 min for complete dehydration. A negative photoresist SU-8 2050 (Micro-Chem Corp, Newton, MA, USA) was spin-coated on top of each wafer using a spin coater (CEE-200; Brewer Science, Rolla, MO, USA). The wafers were then soft-baked at 65 °C for 5 min and then at 95 °C for 30 min. The wafers were aligned using an infrared mask aligner and exposed to ultra-violet (UV) light using a first mask containing the channels and chambers. The wafers were hard-baked at 65 °C for 5 min and 95 °C for 30 min. To achieve the desired thickness of the channels and chambers and form phase-guide structures, the second set of spin coating, soft bake, UV light exposure, and hard bake was performed for each wafer using a second mask containing the phase-guide structures as well as the channels and chambers. Then, the wafers were rinsed in a SU-8 developer for 17 min to remove the unexposed photoresist, followed by isopropyl alcohol (IPA) rinsing. Finally, the wafers were blown dry with compressed nitrogen air to remove IPA. To protect the mold and facilitate the release of PDMS from the mold, a hydrophobic release agent was coated on the surface by exposing the mold surface to hexamethyldisilazane (HMDS) vapor. After generating the mold structures, the PDMS base and curing agent were mixed at the ratio of 10:1 and then degassed in a vacuum chamber to remove air bubbles. Next, liquid PDMS was poured onto the wafer mold and cured at 80 °C for 3 h. After cooling, the PDMS replicas were peeled from the mold, and the inlet and vent ports were punched using punch biopsy. The bottom PDMS and a glass slide substrate were treated with oxygen plasma and then aligned and placed in contact for a permanent bond. The top PDMS was aligned and placed in contact with a bottom PDMS to form the device.

### 3.2. Sample Preparation for qPCR

Keratinocytes were isolated from neonatal (1 to 3-day-old neonates) foreskin tissues obtained from John R. Oishei Children’s Hospital, Buffalo, NY, in accordance with the guidelines of this IRB of John R. Oishei Children’s Hospital. Skin samples were washed three times with phosphate-buffered saline (PBS), cut into small pieces (0.5 cm × 0.5 cm), and enzymatically digested with dispase II protease (Sigma, St. Louis, MO, USA) at 4 °C for 12–18 h. Using fine forceps, the epidermis was separated from the dermis and further treated with trypsin-EDTA (0.25%) (Life Technologies, Carlsbad, CA, USA) at 37 °C for about 10–15 min. The cell suspension was filtered through a 70-μm cell strainer (BD Biosciences, Franklin Lakes, NJ, USA), centrifuged, and seeded on a confluent monolayer of growth-arrested 3T3-J2 mouse fibroblast feeder cells (ATCC; Manassas, VA, USA) using keratinocyte growth medium (KCM) media, which contains a 3:1 mixture of DMEM (high glucose) and Ham’s F-12 medium (Life Technologies, Carlsbad, CA, USA) supplemented with 10% (*v/v*) fetal bovine serum (FBS; Atlanta Biologicals, Flowery Branch, GA, USA), 100 nM cholera toxin (Vibrio Cholerae, Type Inaba 569 B; Millipore, Burlington, MA, USA), 5 μg/mL transferrin (Life Technologies, Carlsbad, CA, USA), 0.4 μg/mL hydrocortisone (Sigma, St. Louis, MO, USA), 0.13 U/mL insulin (Sigma, St. Louis, MO, USA), $1.4\times 10^{-4}$ M adenine (Sigma, St. Louis, MO, USA), $2\times 10^{-9}$  M triiodo-L-thyronine (Sigma, St. Louis, MO, USA), 1× antibiotic-antimycotic (Life Technologies, Carlsbad, CA, USA), and 10 ng/mL epidermal growth factor (EGF, added after 3 days of seeding; BD Biosciences, Franklin Lakes, NJ, USA). After 7–10 days of the culture, feeder cells were removed using versene treatment and keratinocytes were harvested using trypsin-EDTA (0.25%); neutralized with PBS containing 10% FBS, and further cultured on collagen type I-coated tissue culture plates (10 μg collagen type I per ${\mathrm{cm}}^{2}$; BD Biosciences, Franklin Lakes, NJ, USA) in keratinocyte serum-free growth medium (KSFM, Epilife medium with human keratinocyte growth supplement; Life Technologies, Carlsbad, CA, USA).

### 3.3. Performance of qPCR

Total RNA was isolated from keratinocytes using the RNeasy kit (Qiagen, Germantown, MD, USA) according to the manufacturer’s instructions. Subsequently, cDNA was synthesized from total RNA using QuantiTect reverse transcription kit (Qiagen, Germantown, MD, USA). First, a microfluidic device was cleaned with RNase Away (Thermo Scientific, Waltham, MA, USA) to wash out ribonuclease, which catalyzes RNA degradation, followed by cleaning with nuclease-free water to remove the residue of RNase Away. Then, the cDNA sample was loaded into inlet A and nuclease-free water into inlet B. After completing the serial dilution in a device using the process shown in Figure 3, diluted samples were collected from each dilution chamber (#1D through #5D) using fine needles. The performance of our proposed device was verified by qPCR using those diluted samples. Amplification of the diluted cDNA samples was performed using a real-time PCR detection system (CFX96; Bio-Rad, Hercules, CA, USA) with the SYBR green mix (Invitrogen, Carlsbad, CA, USA) according to the manufacturer’s instructions. qPCR is a simple and powerful method allowing in vitro amplification of DNA fragments through a succession of three incubation steps at different temperatures [45,46]. In this study, a hot start (600 s at 95 °C) was followed by 40 cycles of denaturing (15 s at 95 °C), annealing (30 s at 60 °C), and extension (30 s at 60 °C). In fact, using this ten-fold serial dilution, we can study the expression of RPL32 over high dynamic range ten-fold serial dilutions; however, this was not the main focus of this study. After qPCR cycling, the specificity of each product was verified using melt curve analysis, which is frequently used as a diagnostic tool for assessing qPCR assays. For manually diluted samples, cDNA was diluted using 1 µL of cDNA along with 9 µL of nuclease-free water to achieve the same ten-fold serial dilution in each step.

## 4. Results

### 4.1. Volume Analysis

After completing the operating process, samples were expected to have the same amount of volume in each dilution chamber with about 8 μL. However, the measured volumes showed a slight deviation, as shown in Figure 6a. One possible reason was that the loaded sample volumes in each dilution chamber were not as accurate as of the target values. Right after the step shown in Figure 4iii, the liquids loaded in each dilution chamber were retrieved by fine needles and measured using adjustable volume micropipette aspiration in triplicate, as shown in Figure 6b. The errors may partly arise when the liquids adhere to the chamber edge of the capillary valve between the metering chamber and the dilution chamber (Appendix A). Here, for clarification, more than four test devices underwent step (i) to step (viii), and sample volume was measured for Figure 6a. More than another four sets of the test devices underwent step (i) to (iii), and sample volume was measured for Figure 6b. Additionally, non-uniform chamber thickness over the device area related to microfabrication errors may be a possible factor underlying the observed deviation. However, the errors were still acceptable for performing qPCR as proof-of-concept verification.

### 4.2. Proof-of-Concept: Verification of the Performance of the Proposed Serial Dilution by qPCR

Although fluorescent analysis is widely used for serial dilution analysis, there are limitations in detecting a wide range of concentrations due to background noises [2,47]. As an alternative, a qPCR assay can be used to verify the performance of the serial dilution because it provides an unusually large dynamic range across several orders of magnitude. For proof of concept, in this study, qPCR was performed after preparing a series of ten-fold serial dilutions of the cDNA sample using a device as previously described, as well as by manual pipetting. The measurement of each concentration point was performed in triplicate. Figure 7 shows plots of the measured cycle quantification values, $C_{q}$, against the concentrations of serially diluted cDNA sample obtained by using the proposed microfluidic device and manual pipetting. Here, $C_{q}$ is a PCR thermal cycle number at which the reaction curve intersects with a threshold line, such as background fluorescence [45], and the $C_{q}$ values show a linear relationship with the sample concentration across several orders of magnitude. The result showed a strong linear relationship between the $C_{q}$ values and the ten-fold serial dilutions of cDNA sample ranging from $10^{0}$ to $10^{-4}$, with $R^{2}>0.99$ obtained using the line of best-fit, which was equivalent to that for the cDNA sample prepared by manual pipetting. The $C_{q}$ slope of the microfluidic results is not as sharp as the one from the manual results. Some of the possible reasons are that; the volume of cDNA and buffer were not accurate (e.g., larger) compared with what we expected in this qPCR test as well, considering the volume analysis results in Figure 5, insufficient mixing, etc. resulting the deviation. To ensure the validity, repeating the qPCR tests to add error bars are needed in the future study.

### 4.3. Durability, Limitations and Future Studies

Because the liquids were partially driven by gas-solubility-based vacuum-driven pumping, the degassed PDMS without additional coating would lose pumping ability after a certain time. As reported by several papers [11,13,48,49], the flow rate using the degassed PDMS decreases after the device is exposed to the atmosphere. The characteristic time, *τ*, can be estimated by Fick’s law [12]:(6)$$\tau \approx L^{2}D^{-1},$$
where *L* is the length between the top surface of the chamber and the outer surface of the PDMS device and *D* is the diffusion coefficient of air in PDMS. Therefore, the diminishing characteristic time, *τ*, depends on the geometry of the device (e.g., the length, thickness, or volume of the PDMS device, and the surface area exposed to the atmosphere allowing air diffusion) and the property of the PDMS material (e.g., the mixing ratio of PDMS base and curing agent [50]). For instance, if L=5 mm and $D=3.4\times 10^{-9}{\;m}^{2}{\;s}^{-1}$, the diminishing characteristic time would be τ≈ 2 h. Currently, the total operation time for the ten-fold serial dilution with 7 to 10 μL in each chamber was 45 min. Possible approaches to increase the ability of steady vacuum-driven pumping over a longer period of time are to coat the outer surface of the PDMS device with sealing materials (e.g., wax [48] or tape [13]) and to construct a thicker PDMS device [11,13]. The total operation time seems to be restricted by relatively slow vacuum-driven pumping (e.g., a few ${\mathrm{nL}\;s}^{-1}$) [11,13]. During the metering process, more than 5 μL of excess liquid buffer was driven to the buffer waste chamber with close to 10 ${\mathrm{nL}\;s}^{-1}$, which would decay quickly over time. Additionally, about 0.9 μL of the liquid sample was transferred across the transfer bridge, with even slower flow rates, four times. In the future, in order to reduce the total operation time, the metering process will be replaced by centrifugal pumping rather than vacuum-driven pumping [2,3]. Using tapes to seal the inlet and vent ports is inconvenient; this would be resolved by incorporating active or passive valves with smart sealing functions for vacuum-driven pumping. In the test, the trapped sample in the capillary bridges was used to assist in forming a dead-end channel. The trapped residues were targeted to be controlled by, such as employing small pockets at the edges of the metering chambers and applying slightly higher RPM than critical burst frequency. However, improving the robustness is required, and in the future, we plan to implement valves instead. Applying high RPM on a device is a challenge, as a device is made from a thick PDMS slab, and the substrate is not a round shape in the experiment. A robust symmetry platform will be developed in a future study. The concept of the proposed device can be used to achieve different dilution ratios by altering its geometry (e.g., volume of the metering chambers, phase-guide and the dilution chambers). Here, achieving the desired thickness is a necessary in fabrication by avoiding the edge beads. Furthermore, we would integrate an on-chip qPCR module with the serial dilution module by devising a method to fill the empty space in each chamber with oils to avoid evaporation and contamination during on-chip thermal cycling.

## 5. Conclusions

In conclusion, we propose a new serial dilution method utilizing centrifugal and vacuum-driven force. The combination of two actuation methods could complement the drawbacks of each. A qPCR assay was carried out to verify the performance of the on-chip serial dilution, showing strong linearity over the large dynamic range across several orders of magnitude. Even though further improvements are needed for accurate volume control, reduction in the total operation time, a user-friendly sealing method, and additional on-chip integrations, the proposed device demonstrates feasibility for promising on-chip serial dilution applications. Furthermore, in addition to centrifugal pumping, this vacuum-driven pumping could easily be integrated with other types of pumping for applications relating to various novel on-chip biological or chemical analyses in a hybrid manner.

## Figures and Tables

**Figure 1 micromachines-12-00482-f001:**
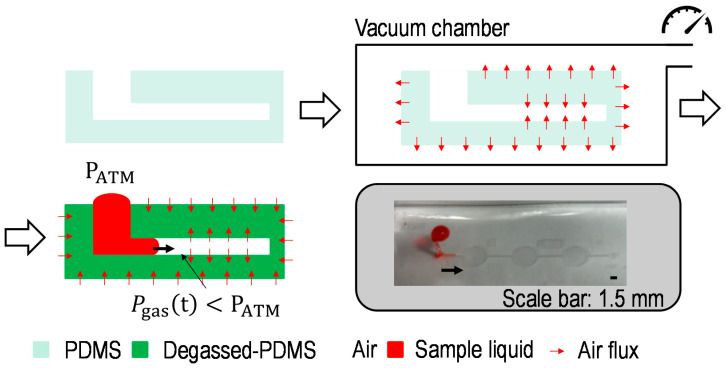
General device configurations and preparation for the degassed PDMS-based vacuum-driven pumping. When the whole microfluidic device is made of materials (e.g., PDMS), it can function as a self-powered vacuum source. The device is pre-degassed before usage to let dissolved air is driven out. Once the degassed device is exposed to air again, the materials start to reabsorb air from all surroundings, including embedded microfluidic channels, resulting in a vacuum that will draw sample liquids along the microfluidic channels. The image shows a simple microfluidic device to demonstrate the vacuum-driven pumping using the degassed PDMS.

**Figure 2 micromachines-12-00482-f002:**
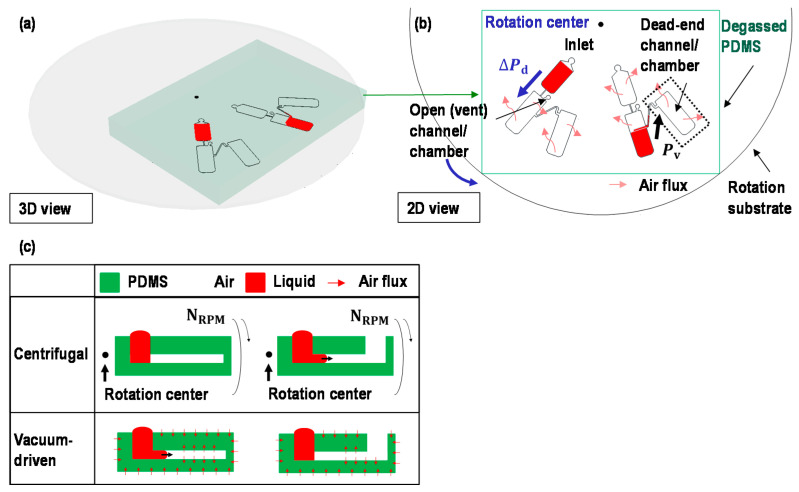
(**a**) The 3D and (**b**) 2D schematics of the operation principle using driving pressure by centrifugal force, $\Delta P_{d}$, or vacuum-driven pressure, $P_{v}$. The centrifugal pressure allows the sample to be driven in a one-way direction, while the vacuum-driven force can be used for various directions to drag the sample without any rotation. To switch the dominant forces between centrifugal force and vacuum-driven force to drive samples, the fluidic passage should be wisely designed. For instance, if we let the liquid sample go through the open chamber and the closed chamber one after another, we can make the situation to let the sample driven by centrifugal force (with a rotation) and the vacuum-driven force one after another. (**c**) The comparison between the centrifugal flows and the vacuum-driven flows in each dead-end and open-end channel. The centrifugal pressure requires an open-end channel to drive a sample, while the vacuum-driven pressure requires a dead-end channel to drive a sample. In the proposed device, a dominant force is switched between the centrifugal force and the vacuum-driven force. $N_{\mathrm{RPM}}$ is a rotational speed.

**Figure 3 micromachines-12-00482-f003:**
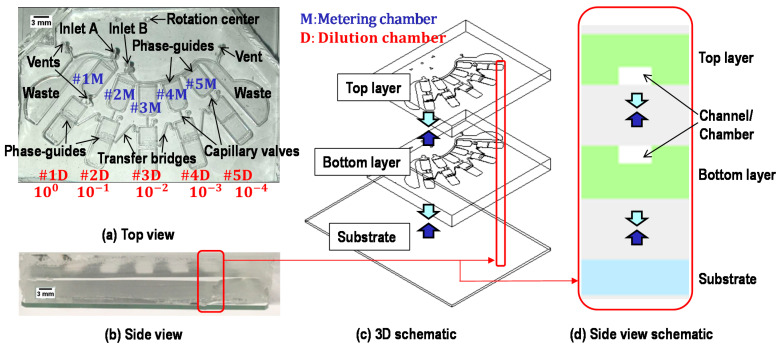
Device images and schematics. Metering chambers corresponding to serial dilutions with $10^{0},\;10^{-1},10^{-2},\;10^{-3}$, and $10^{-4}$ concentrations are labeled as #1M, #2M, #3M, #4M, and #5M, respectively. Dilution chambers corresponding to serial dilutions with $10^{0},10^{-1},10^{-2},10^{-3}$, and $10^{-4}$ concentrations are labeled as #1D, #2D, #3D, #4D, and #5D, respectively. There are three vents in total; two vents in the waste chambers are used for sample metering, and one vent in #1D is used for sample transfer.

**Figure 4 micromachines-12-00482-f004:**
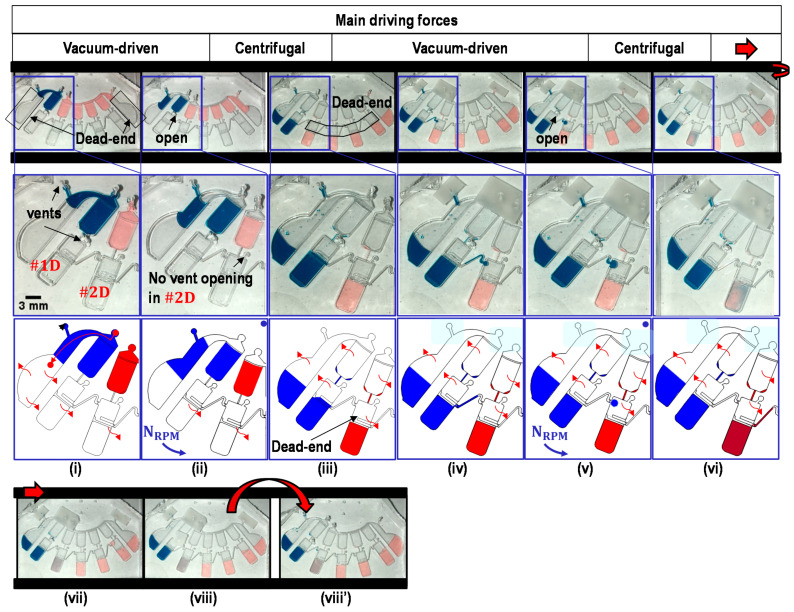
Sequential images of the serial dilution with liquid sample (water containing blue dye) and liquid buffer (water containing red dye): (**i**) liquid loading; (**ii**) liquid metering; (**iii**) liquid transfer from metering chambers into dilution chambers; (**iv**) partial transfer of one-tenth of its original volume from #1D to #2D, where inlets and vents were sealed with adhesive tape, except one vent in #1D; (**v**,**vi**) mixing of the liquid sample and the liquid buffer; (**vii**) repeating of the serial dilution process, and (**viii**,**viii’**) completed serial dilution showing the device with and without tapes. Here, for clarification, vent ports design located in #2D to #5D were not punched in this study.

**Figure 5 micromachines-12-00482-f005:**
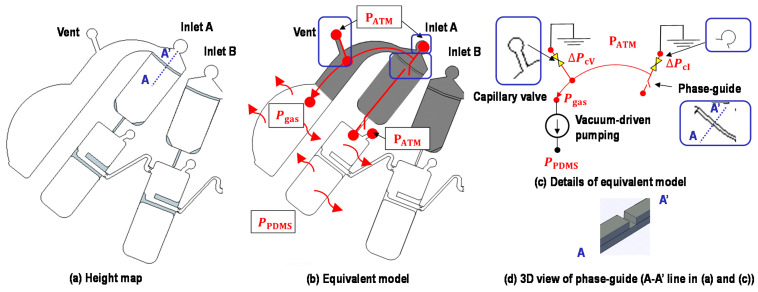
A height map and equivalent model for the vacuum-driven metering process. (**a**) Colored parts were designed to have 210 μm, while non-colored parts were designed to have 420 μm. Along with lateral geometrical confinements, the height differences assist in determining the fluidic passage (such as liquid sample stop, split, confinement, etc.). (**b**) Equivalent model and (**c**) details of equivalent model. Colored parts represent the liquid filling parts before the metering. The pressure difference across the liquid column is generated due to the gas solubility of degassed PDMS, driving liquids into the dead-end channel/chamber. Geometrical confinement by phase-guide structures would play a role in metering the liquids. $P_{\mathrm{ATM}}$ is the atmospheric pressure, $P_{\mathrm{gas}}$ is the pressure inside the chamber, $P_{\mathrm{PDMS}}$ is the pressure associated with degassed PDMS, and $\Delta P_{\mathrm{cV}}$ and $\Delta P_{\mathrm{cI}}$ are the capillary pressure of the capillary valves at the vent and the inlet, respectively. (**d**) 3D schematic view of phase-guide for one side of a PDMS layer.

**Figure 6 micromachines-12-00482-f006:**
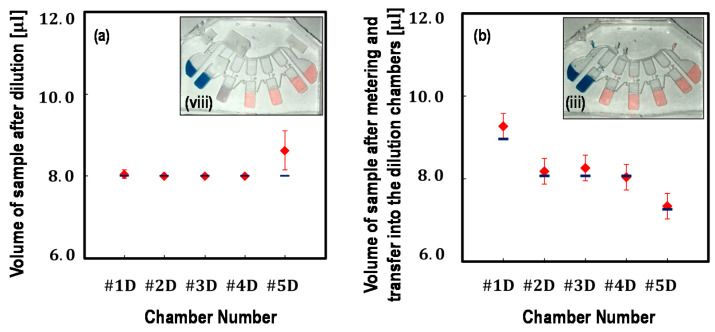
Volume analysis: (**a**) the measured volumes of liquids in each chamber after serial dilution (shown in Figure 4viii,viii’) and (**b**) the measured volumes of liquids in each chamber after metering and transfer into each dilution chamber (as shown in Figure 4iii). Digital micropipette aspiration was used to measure the sample volume. More than four test devices were used for measuring for (**a**) and (**b**), respectively. The insets indicate in which process the samples were collected for measuring.

**Figure 7 micromachines-12-00482-f007:**
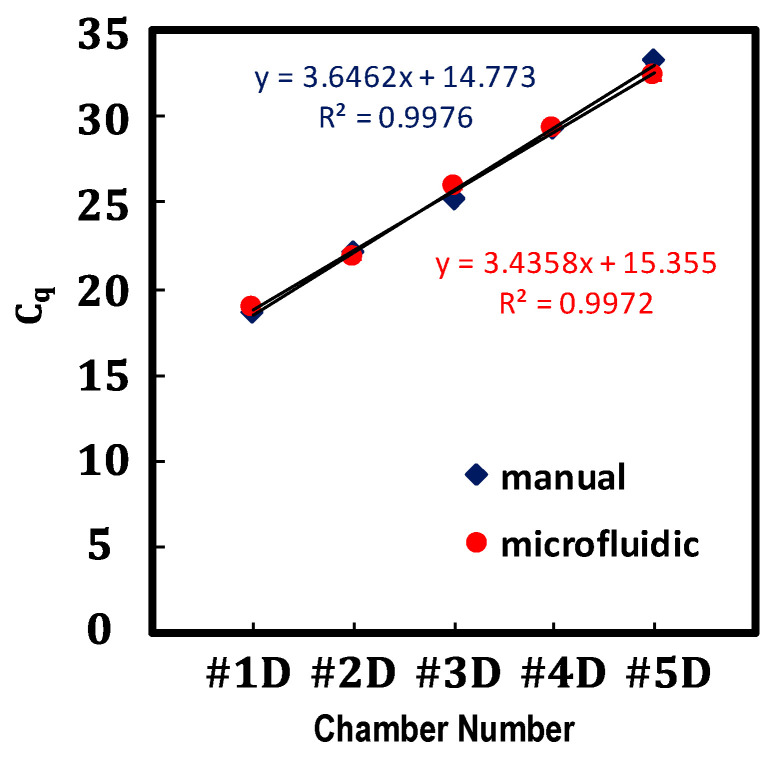
Plots of the measured qPCR cycle quantification values, $C_{q}$, against the concentrations of serially diluted cDNA samples ranging from $10^{0}$ to $10^{-4}$, obtained by using the proposed microfluidic device and manual pipetting.

## Data Availability

Data are available upon contact.

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
