# Peer review of "An Integrated Centrifugal Degassed PDMS-Based Microfluidic Device for Serial Dilution"

_micromachines, 2021, doi:10.3390/mi12050482_

Round 1
Reviewer 1 Report
The paper explains an innovative microfluidic approach for serial dilution work which is an important work for many applications in the field. The approach utilize flow and pressure difference to work on dilution that can work up to 5 chambers concurrently.
The work surely will gain much attention to the reader of the field and it will also provide a good alternative or solution to researchers who need such technique, particularly in biomedical application.
However, there are a few small improvement that can be made for this manuscript as following:
- Figure 1, 3D view can be either enlarged or improved in terms of its dpi. The current resolution is not clear when it is zoom in.
- Figure 2, Top view needs improvement in terms of its arrangement. The caption for both #1M or #1D are rather confusing. Proper arrangement will make the figure stand out itself.
- The current description on result and discussion are great but the pressure difference profile between chamber are not described. This might cause a potential application limitation. Though centrifugal force will be the main driving force to get the device flow from 1st to last chamber, it will be great if the pressure difference for each chamber can be either measured experimentally or numerically to gauge the understanding of the flow physics.
Author Response
Thank you for your letter and for the reviewers’ comments concerning our manuscript entitled ‘An integrated centrifugal degassed PDMS-based microfluidic device for serial dilution’ (micromachines-1190406). Those comments are all valuable and very helpful for revising and improving our paper, as well as the important guiding significance to our researches. We have studied comments carefully and have made correction which we hope meet with approval. Please find the attached pdf file for the details as well as the revised manuscript.

Reviewer 2 Report
- The authors report a serial dilution method utilizing the combination of centrifugal and 470 vacuum-driven force.
- No doubt the method and device have lack no innovation merit, however, I found the paper hard to read. The authors need to explain the various concepts used in the paper (e.g. phase guide, degassed PDMS, etc) despite they cite relevant literature, but brief description is necessary. In addition, despite the paper presenting a proof of concept of a technically sound method, the results are minimally interpreted and discussed (e.g. Fig.6). The results that support the demonstration of serial dilution are not sufficient. For example, to support the results in Fig.6, fluorescent imaging data might be included.
- Authors need to discuss the role of degassed PDMS and how its utilized for pumping. Schematic drawing might be included.
- 1 is not sufficiently explaining the concept. Need improvement
Lines 45-56: A typical 45 procedure involves manual pipetting of samples into each micro well and mixing with 46 dilution buffers. This is a time-consuming process … The author may also talk about some of the available technologies which become routinely used (e.g. Tecan system).
Line 139: “The dominant force is switched between the centrifugal force and the vacuum-driven force during device operation.” How?
Lines 152-153: “metering 152 chambers are designed to allocate 8.96 μl in #1M; 8.07 μl each in #2M, #3M, and #4M; and 153 7.26 μl in #5M” Why?
Lines 188-190: “These trapped samples partially 188 contribute to form a dead-end chamber, which can trigger vacuum-driven force to convey 189 the sample in the later process” This is not clear. Please show in figure.
In section 2.2: How to ensure that one-tenth of the sample transferred from 1D to 2D? How this was optimized?
Fig.4: Inlets A and B need to be indicated in Fig. 4a.
Lines 216-218: “Here, the strong constraint at the phase-guide structure causes the liquid column to sepa-216 rate into two parts: one flowing into the waste chamber and the other retained inside the 217 metering chamber” What is the exact mechanism?
Line 226: What is Δ?cI?
Lines 268-270: “Here, the phase-guide structures were designed to meter and transfer one-tenth of its original volume from the previous dilution chamber into the next dilution chamber.” Please explain how?
Page 10: What happened in Fig.5b? Pls explain. Fig. 5 should be under the results section (?).
Author Response

(The authors gave the same response as above.)

Reviewer 3 Report
The manuscript, “An integrated centrifugal degassed PDMS-based microfluidic device for serial dilution” by Anyang Wang et al, describes PDMS based microfluidic devices with many functional components including metering chambers, concentration chambers, inlet and venting ports, capillary valves, etc. to demonstrate the capability of precise transporting and diluting capabilities by combination of vacuum and centrifugal forces. The manuscript should get published eventually after addressing all my comments:
- How vacuum was maintained during operation? After the fluidic sample were taken from the vacuum oven/initial loading liquid step, the chamber should be put in Patm as well. What’s the special steps to maintenance the vacuum? Will the vacuum chamber deform under different pressure loadings inside and outside?
- How to remove the diluted samples after each operation?
- What’s the overall time length for each step and the how dilution process? There are many steps involving manual valve operation, fluid transferring and centrifugal steps, which leaves an impression of slow operation using this device.
- In Fig. 3, I noticed there were PDMS patches added to the port in step (iv) ~ (vi). Are the devices different between step (i)-(iii) and (iv)-(vi)?
- It is a little confusing for Fig. 4 (a) in terms of concentration chambers. It looked like the chamber was divided by upper and lower half by the centrifugal capillary channel. How does this part designed and it is recommended to mark different function parts with different colors.
- In Fig. 2 (d), are the top and bottom channel sealed or open after peeling off from the Si wafer. It looked the channel is sealed from the drawing.
Author Response

(The authors gave the same response as above.)

Reviewer 4 Report
The authors describe a serial dilution generator combining centrifugal and vacuum-driven forces. The physical phenomena underlying the device functionality are well described and the device performance are properly characterized. A proof-of concept validation of the device was performed using qPCR as a read-out. I appreciate the critical discussion about the limitation of the system as well as the description of potential future improvements.
For these reasons, I strongly recommend its publication in the present form.
Author Response
Thank you for your letter and for the reviewers’ comments concerning our manuscript entitled ‘An integrated centrifugal degassed PDMS-based microfluidic device for serial dilution’ (micromachines-1190406). Those comments are all valuable and very helpful for revising and improving our paper, as well as the important guiding significance to our future researches.
Round 2
Reviewer 2 Report
The manuscript has been significantly improved and can be accepted.